# Prevalence and Diversity of Blood Parasites (*Plasmodium*, *Leucocytozoon* and *Trypanosoma*) in Backyard Chickens (*Gallus gallus domesticus*) Raised in Southern Thailand

**DOI:** 10.3390/ani13172798

**Published:** 2023-09-03

**Authors:** Kanpapat Boonchuay, Thotsapol Thomrongsuwannakij, Carolina Romeiro Fernandes Chagas, Pornchai Pornpanom

**Affiliations:** 1Akkhraratchakumari Veterinary College, Walailak University, Nakhon Si Thammarat 80160, Thailand; kanpapat.bo@wu.ac.th (K.B.); thotsapol.th@wu.ac.th (T.T.); 2Nature Research Centre, Akademijos 2, 08412 Vilnius, Lithuania; crfchagas@gmail.com; 3Informatics Innovation Center of Excellence, Walailak University, Nakhon Si Thammarat 80160, Thailand; 4One Health Research Center, Walailak University, Nakhon Si Thammarat 80160, Thailand

**Keywords:** backyard chicken, Leucocytozoon, *Plasmodium*, *Trypanosoma*, Thailand

## Abstract

**Simple Summary:**

Chickens can be infected by several avian blood parasites (*Plasmodium*, *Haemoproteus*, *Trypanosoma* and microfilaria) that can cause a big impact on poultry production. However, some of them are known to cause a high impact on poultry production (*Plasmodium* and *Leucocytozoon*), while others still require further investigation. Raising backyard chickens is a common practice in Thailand, and the low-biosecurity system in which they are kept favors the transmission of vector-borne diseases, which include several blood parasites. The spread of such infections can compromise production, resulting in economic impact. This study aimed to report the molecular prevalence, lineage diversity and morphology of blood parasites infecting backyard chickens in three different provinces in Thailand. We found a high prevalence of *Plasmodium* sp. and *Leucocytozoon* sp. infections, while Trypanosoma and microfilaria had a lower prevalence. *Plasmodium gallinaceum* and *Leucocytozoon macleani* were present in the studied individuals as well as Trypanosoma, which resembles *T. calmettei*. The buffy-coat method and molecular analysis were shown to be valuable diagnostic tools for blood parasites in chickens. These results can be used to promote awareness of parasite infections in the study area.

**Abstract:**

Avian malaria and leucocytozoonosis can cause fatal diseases, whereas avian trypanosomiasis is reported to be harmless in chickens. Backyard chickens can be infected by several pathogens, including blood parasites, that may shed to industrial poultry production, with a consequently higher economic impact. This study aimed to investigate the presence of several blood parasites (*Plasmodium*, *Leucocytozoon* and *Trypanosoma*) in backyard chickens raised in Southern Thailand, using PCR-based detection and microscopic methods. From June 2021 to June 2022, 57 backyard chickens were sampled. Fresh thin blood smears were prepared from 11 individuals, and buffy coat smears were prepared from 55 of them. Both thin blood smears and buffy coat smears were used for microscopic analysis. Two nested PCR protocols that amplify a fragment of cytochrome *b* (*cytb*) and small subunit rRNA (*SSU* rRNA) genes were used to identify Haemosporida and *Trypanosoma* parasites, respectively. The number of positive samples was higher with the application of nested PCR than when buffy coat smears were used. Three new *Plasmodium* lineages (GALLUS47-49) and thirteen *Leucocytozoon* lineages (GALLUS50-62) were found. Trophozoites, meronts and gametocytes of *Plasmodium gallinaceum* (GALLUS01) were present in one thin blood smear. All thin blood smears revealed *Leucocytozoon* infections, but only three samples were a single infection. These three samples revealed the presence of fusiform host cell–parasite complexes, of which the morphological features resembled those of *Leucocytozoon macleani* (possible synonym is *Leucocytozoon sabrazesi*), while the *cytb* showed that this parasite is closely related to the lineage GALLUS06-07, described as *Leucocytozoon schouteni*. The *Trypanosoma* prevalence was 33.33%; it was present in only one of the thin blood smears, and it resembles *Trypanosoma calmettei*. This study showed the prevalence of a high diversity of *Plasmodium* (64.91%) and *Leucocytozoon* (89.47%) in Thai chickens. Both nested-PCR and buffy coat smear can be used as the diagnostic tool for the testing of *Plasmodium*, *Leucocytozoon* and *Trypanosoma* for parasitic control in backyard chickens and poultry farms. The information on the parasite species that can be found in chickens raised in Southern Thailand was also considered as the baseline information for further study.

## 1. Introduction

Avian haemosporidian (Apicomplexa: Haemosporida) includes the genera *Haemoproteus*, *Plasmodium*, *Fallisia* and *Leucocytozoon*, and these are vector-borne parasites distributed worldwide, except Antarctica [1,2,3]. There are about 277 described species among 23 host orders [4,5,6,7]. In domestic chickens (*Gallus gallus domesticus*), two *Plasmodium* species were described: *Plasmodium gallinaceum*, likely endemic in Asia, and *Plasmodium juxtanucleare* [4], which has a global distribution [8]. Three *Leucocytozoon* species were reported in domestic chickens: *Leucocytozoon macleani* (possible synonym is *Leucocytozoon sabrazesi*), *Leucocytozoon schouteni* and *Leucocytozoon caulleryi* [4]. The *Trypanosoma* spp. found in domestic chickens include *Trypanosoma calmettei*, *Trypanosoma gallinarum* and *Trypanosoma numidae* [9].

These parasites are transmitted by various vectors. *Plasmodium gallinaceum* and *P. juxtanucleare* are transmitted by mosquitoes (Diptera: Culicidae), with *P. gallinaceum* being transmitted mainly by *Mansonia crassipes* and *Culex quinquefasciatus* [4,10], while *P. juxtanucleare* is mainly transmitted by several species of *Culex* mosquitoes [4]. *Leucocytozoon* spp. are transmitted by several species of back flies (Diptera: Simuliidae), but only *L. caulleryi* is transmitted by insects belonging to the *Culicoides* genus (Diptera: Ceratopogonidae) [11,12]. For *Trypanosoma* spp., these parasites are transmitted by blood-sucking arthropods belonging to families Simullidae, Culicidae, Ceratopogonidae, Hippoboscidae (Diptera) and Dermanyssidae (Mesostigmata) [13,14,15,16].

Generally, the clinical signs of *Plasmodium* infection in chickens (known as avian malaria) are greenish feces, anemia, depression, reduced weight gain, fluffed-out feathers and often death [17]. *Leucocytozoon* causes a malaria-like disease called leucocytozoonosis [18,19], and its pathogenicity can vary according to the vertebrate host, parasite species and lineage [20,21,22,23]. This important disease was first reported in Vietnam in 1909 [18]. *Leucocytozoon caulleryi* can cause a lethal hemorrhagic disease [24,25], and infected chickens frequently exhibit anemia, anorexia, ataxia, lethargy, green diarrhea, pallor, decreased egg production and eventually death [9,25]. *Leucocytozoon macleani* and *L. schouteni* are less pathogenic but can still cause anemia and greenish droppings, slight emaciation and reduced egg production [4,9]. Avian trypanosomes have been reported as non-pathogenic for domestic chickens, due to their impact in wild birds being rarely reported and understudied [26,27]. Even though about 100 species of avian trypanosomes have been described, they are poorly known [13,28,29].

Backyard chickens is one of the common low biosecurity poultry production systems in Thailand [30]. The animals raised in this system are not kept in areas protected from blood-sucking insects that can transmit these parasites. They can easily be bitten by those insects that can be infected by blood parasites (*Plasmodium* sp., *Leucocytozoon* sp. and *Trypanosoma* sp.) which will be injected during the blood meal. Once infected, these backyard chickens can act as reservoirs of these pathogens that can shed to other households or the bigger sector of poultry production [31], resulting in serious economic impact. A combination of microscopic and PCR-based methods is helpful for the diagnosis of avian blood parasites and can show not only whether the infected animal is a reservoir of avian blood parasites (when the parasite can be seen in the blood) but also the species and lineages (haplotypes) that are circulating in the region [32]. Therefore, this study aimed to investigate the presence of *Plasmodium*, *Leucocytozoon* and *Trypanosoma* in backyard chickens raised in Southern Thailand. We also report microscopic and molecular characteristics of parasites found in backyard chickens. This information might be helpful for parasite prevention, parasite elimination and further studies.

## 2. Materials and Method

### 2.1. Sample Collection and Processing and Microscopic Examination

Blood samples were collected during a one-year period, between June 2021 to June 2022. This allowed us to collect samples during the rainy season (June–September) and dry season (April–May). A maximum 1 mL volume of blood was taken from the brachial vein of 57 backyard chickens raised in Nakhon Si Thammarat (8°25′ N, 99°58′ E), Phatthalung (7°37′ N, 100°4′ E) and Surat Thani (9°7′ N, 99°20′ E). The blood was immediately transferred to tubes containing EDTA as an anticoagulant (MediPlus^TM^, Bangkok, Thailand) and transported to the Laboratory of Hematology, Akkhararatchakumari Veterinary College, Walailak University, where the analyses were processed. During the transport, these samples were kept in icebox until processing. From eleven samples, two to five fresh thin blood smears (without any anticoagulant) were prepared using a few drops of blood (all from native chickens) and dried using an electric fan for further microscopic analysis. 

In the laboratory, samples stored in EDTA-tubes were used to prepare buffy coat smears according to Chagas et al. [33] with some modifications. Briefly, EDTA-blood was filled into a microhematocrit capillary tube and centrifuged at 12,000 rpm for 5 min. Then, buffy coat layer was transferred into a glass slide and smeared, allowed to air dry at room temperature, fixed in absolute methanol for one minute and stained with a 10% Giemsa solution for 45 min. The remaining samples in the EDTA-tube were frozen for further molecular analysis, which is described below. The freshly prepared thin blood smears were fixed and stained as indicated for the buffy coat smear.

Microscopic examination for both buffy coat smear (55 samples) and fresh thin blood smears (11 samples) were conducted following the previous report [32]. Briefly, 100 fields were examined at 400× as well as 100 fields at 1000× magnification. Parasitemia was calculated for the samples with thin blood smears available as a percentage of the infected cells per 10,000 red blood cells [34]. A light microscope was used for the examination of blood smears and for collecting images of blood parasites. A microscope Nikon ECLIPSE C*i*-L (Nikon, Tokyo, Japan) equipped with a Nikon DS-F*i*3 digital camera (Nikon, Tokyo, Japan) together with NIS Elements D imaging software (version 5.01, Nikon, Tokyo, Japan) was used for analysis of buffy coat smears. Blood smears were analyzed using an Olympus BX43 (Olympus, Tokyo, Japan) equipped with an Olympus DP27 digital camera (Olympus, Tokyo, Japan) together with CellSens imaging software (version 1.18, Olympus, Tokyo, Japan).

### 2.2. DNA Extraction, Nested-PCR and Sequencing

Fifty microliters of each blood sample were used for genomic DNA extraction using the Blood Genomic DNA Extraction Mini Kit (FavorPrep, Pingtung, Taiwan) following the manufacturer’s instructions. A DNA fragment of approximately 479 bp of the *cytb* gene of Haemosporida parasites and a 770 bp of the small subunit ribosomal RNA (*SSU* rRNA) of *Trypanosoma* parasites were amplified using two different nested-PCR protocols with some modification of thermal profile [13,35,36]. In brief, the PCR mix for all parasites was prepared in a total volume of 20 µL containing 10 µL of PCR master mix (OnePCR^TM^ Ultra, Bio-Helix, New Taipei City, Taiwan), 1 µL of each primer (concentration = 10 µM), 6 µL of water and 2 µL of DNA template (concentration < 25 ng/µL in most samples). A negative (ultra-pure water) and several positive controls—*Plasmodium* sp. *cytb* lineage GLACUC08 [37], *Leucocytozoon* isolate SEO-KU483 [38] and *Trypanosoma* isolate CSO-KU127 [29]—were used in every run. Thermal profile of nested-PCR amplifying *cytb* and *SSU* rRNA was followed [38,39]. The amplicons were checked using 1.5% agarose gel prepared from the Agarose Tablets (Bio-Helix, New Taipei City, Taiwan). The amplicons were then submitted to the U2Bio Thailand (Bangkok, Thailand) for gel extraction, DNA purification and Sanger sequencing for both forward and reverse strands.

### 2.3. Sequence Analysis and Phylogenetics

*Plasmodium*, *Leucocytozoon* and *Trypanosoma* sequences were evaluated using BioEdit version 7.0.5.3 [40]. Forward and reverse strands were aligned and checked to obtain a contig sequence. Sequences were also checked for the presence of co-infections (presence of double peaks in the electropherograms). If such co-infections were seen, they were excluded from the analysis [41,42]. *Plasmodium* and *Leucocytozoon* sequences from single infections were compared with the sequence deposited on the MalAvi database [43], using BLAST tool. The sequences showing at least one nucleotide of difference were considered as a new lineage [44,45], and it was named according to MalAvi database [43]; all sequences were deposited in MalAvi and GenBank databases (https://www.ncbi.nlm.nih.gov/nucleotide/, accessed on 21 January 2023).

Hemosporidian lineages isolated from our study were used for Bayesian phylogenetic analysis. In total, 30 *Plasmodium* lineages and 39 *Leucocytozoon* lineages were used. Phylogenetic analyses of *Plasmodium* and *Leucocytozoon* were conducted separately. A sequence of *Leucocytozoon* sp. SISKIN2 was used as an outgroup for *Plasmodium* phylogenetic analysis, whereas *Haemoproteus columbae* COLIV03 and *H. iwa* FREMIN01 were used as the outgroup for *Leucocytozoon* phylogenetic analysis. Missing data in each nucleotide position were replaced with “*N*” [37,39].

Bayesian phylogenetic of *Plasmodium* and *Leucocytozoon* were constructed using MrBayes version 3.2.6 [46]. Application of the model of general time-reversible (GTR) was selected with the mrModeltest 2.3 program [47], based on hierarchical likelihood ratio test (hLRT). Markov chain Monte Carlo (MCMC) was run for three million (*Plasmodium*) or five million generations (*Leucocytozoon*), with sampling every 100 generations. The first 25% of three were discarded as “*burn-in*” step. Then, the consensus tree was calculated using the 22,500 and 37,500 remaining trees for *Plasmodium* and *Leucocytozoon*, respectively. The tree was visualized using Figtree version 1.4.3 (http://tree.bio.ed.ac.uk/software/figtree, accessed on 8 November 2022). Jukes–Cantor (JC) model, for which all substitution were equally weighted, was used to calculated the genetic distance between lineages using MEGA version 11 [48].

In total, 49 *Trypanosoma* sequences were used in the phylogenetic analysis, including the isolates from this study. Two sequences of amphibian trypanosomes were used as an outgroup, including *Trypanosoma rotatorium* and *Trypanosoma mega*. The phylogenetics with 1000 replications of bootstrap was constructed using the maximum-likelihood method implemented in MEGA 11 software [48]. Best-fit model was Kimura 2-parameter with gamma distribution (K2 + G), which was selected based on lowest Bayesian information criterion (BIC). Genetic distances were determined using the JC model.

### 2.4. Statistical Analysis

Prevalence of *Plasmodium*, *Leucocytozoon* and *Trypanosoma* infections in Southern Thailand were calculated based on nested-PCR results from the 57 backyard chickens sampled. The confidence intervals (95% CI) were calculated using the function of ‘*binom.approx*’ in package ‘*epitools*’, implemented in R version 4.2.2 [49]. Fisher’s exact test was performed to compare the prevalence of blood parasite infection (based on nested-PCR results) between areas (Nakhon Si Thammarat, Phatthalung and Surat Thani), implemented in R version 4.2.2 [49]. The significance was obtained at *p*-value ≤ 0.05.

## 3. Results

A total of 57 backyard chickens’ (*Gallus gallus domesticus*) blood samples were investigated for the presence of *Plasmodium*, *Leucocytozoon* and *Trypanosoma* infections. This included native chickens (n = 33), hybrid chickens (n = 10), laying hens (n = 5) and fighting roosters (n = 9). These chickens were raised in a backyard production system, with low biosecurity, by local farmer in three provinces located in the central part of Southern Thailand: Nakhon Si Thammarat (NST) where 23 individuals were sampled, Phatthalung (PHL) where 25 individuals were sampled and Surat Thani (SUT) where 9 individuals were sampled. Based on nested-PCR results, 37 samples were positive for *Plasmodium* parasites showing a prevalence of 64.91% (95% CI: 52.52–77.30); 51 were positive for *Leucocytozoon* parasites with a prevalence of 89.47% (95% CI: 81.51–97.44); and 19 samples were positive for *Trypanosoma* parasites showing a prevalence of 33.33% (95% CI: 21.10–45.57) (Table 1). In a comparison of the prevalence of blood parasite infections between areas, the prevalence of *Plasmodium* in NST (69.56%), PHL (32.00%) and SUT (77.77%) were not significantly different (*p* = 0.012); the prevalence of *Leucocytozoon* in NST (78.26%), PHL (96.00%) and PHL (100.00%) were not significantly different (*p* = 0.118); and the prevalence of *Trypanosoma* in NST (0.00%), PHL (60.00%) and SUT (22.22%) were significantly different (*p* < 0.05). 

Of these 37 *Plasmodium*-positive and 51 *Leucocytozoon*-positive samples, 6 *Plasmodium* and 24 *Leucocytozoon* sequences showed double peaks in the electropherogram; considered as co-infections, these sequences were excluded from the sequence and phylogenetic analysis. The remaining sequences, 31 *Plasmodium* and 27 *Leucocytozoon,* were considered as single infections and used in the phylogenetic analysis. Altogether, 6 *Plasmodium* and 16 *Leucocytozoon* lineages were identified in the studied animals (Table 2). 

Microscopic analysis of the buffy coat smears revealed the presence of *Plasmodium* in 5 individuals (Figure 1A–E), *Leucocytozoon* in 48 (Figure 1I,J), and *Trypanosoma* in 4 (Figure 1F–H) (Table 1). Unfortunately, the PCR-protocol used in the study failed to amplify a few microscopically positive samples: two positives for *Plasmodium* and *Trypanosoma* and one sample for *Leucocytozoon*. In the buffy coat smear analyses, we could also observe that a small number of individuals were infected by microfilaria of filarial nematodes (Figure 1L). 

Of the eleven fresh thin blood smear analyses, in one individual (AVC52) the presence of trophozoites, erythrocytic meronts, young and mature gametocytes of *P. gallinaceum* (Figure 2A–F) were verified, the lineage of which was molecularly identified as GALLUS01 (Table 2). The morphological identification was possible due to the presence of the following characteristics: trophozoites were located anywhere in the infected cell (Figure 2A); fully grown erythrocytic meronts markedly displacing the nuclei of erythrocytes occupied more than half of the cytoplasmic space of infected erythrocytes (Figure 2B); growing gametocytes were found in mature erythrocytes, displacing the nuclei of host cells (Figure 2C); and mature gametocytes were varying in shape and markedly deformed host cells and displaced their nuclei laterally (Figure 2D–F). Parasitemia was lower than 0.01%.

In the Bayesian phylogenetic analysis, the five *Plasmodium* lineages identified in the present study were grouped in the *P. juxtanucleare* clade, and they were separated from *P. gallinaceum* clade (Figure 3). In these two clades, *Plasmodium* parasites of Phasianidae birds were separated from avian *Plasmodium* lineages from other host families. The homology in the *P. juxtanucleare* clade was 99.33–100%, whereas in the *P. gallinaceum* it was 100%. *Plasmodium* lineages GALLUS48 and GALLUS49 were closely related to *P. juxtanucleare* GALLUS02, with 99.78% and 99.55% similarity. In contrast, the *Plasmodium* lineage GALLUS47 showed 100% similarity to *Plasmodium* lineage TSUB01. 

All eleven fresh blood smears were positive for *Leucocytozoon* parasites that developed into fusiform host cells–parasite complexes. Of these eleven samples, seven samples were found with only fusiform host cells–parasite complexes, whereas the other four samples were found with both fusiform host cells–parasite complexes and roundish host cells–parasite complexes (Figure 4). Three different genetic lineages were present in the studied samples: GALLUS55, GALLUS61 and GALLUS62 (Table 2). Based on the microscopic analysis of the thin blood smears, the gametocytes of GALLUS55 and GALLUS62 showed fusiform host cells–parasite complexes and roundish host cells–parasite complexes in (Figure 4I,M,K,O), whereas GALLUS61 developed only in fusiform host cells (Figure 4A). These three parasites were suspected to be *L. macleani*. However, the Bayesian phylogenetic inference revealed that these three parasites were grouped in the *Leucocytozoon schouteni* clade (Figure 5), which had the homology ranging between 91.02 and 100%.

One out of eleven fresh thin blood smears showed the presence of tiny trypomastigotes of *Trypanosoma* sp. (individual AVC49). This *Trypanosoma* sp. had a short and slender body (Figure 2G–I). However, due to the presence of only a few trypomastigotes in the sample, it was not possible to morphologically identify it. Unfortunately, the PCR-protocol used failed to amplify the *SSU* rRNA of this parasite. However, some morphometric parameters were measured from five trypomastigotes. Detailed information can be found in Table 3.

It was possible to amplify two *SSU* rRNA sequences of *Trypanosoma* from native chickens (individuals AVC56 and AVC57) raised in Phatthalung. However, due to the lack of fresh thin blood smears, we could not determine their morphological features. These two sequences were used for maximum-likelihood phylogenetic analysis, which revealed that *Trypanosoma* GGD-AVC56 and *Trypanosoma* GGD-AVC57 were phylogenetically grouped in the *Trypanosoma avium* clade (Figure 6). The homology of this clade was 99.74–100%. *Trypanosoma* GGD-AVC56 and *Trypanosoma* GGD-AVC57 had 99.74% similarity to others in this clade. However, our two sequences do not belong to any of the described lineages.

## 4. Discussion

The prevalence of *Plasmodium* spp. and *Leucocytozoon* spp. in backyard chickens in Southern Thailand was high (64.91% and 89.47%, respectively). The study area was located in the central region of southern Thailand that is close to the coastal area, with water bodies that may support mosquitoes and other blood-sucking insects’ development. The examples of insect vector found in Southern Thailand were *Culex* mosquitoes [50], *Culicoides* biting midges [51] and Simuliidae [52]. Additionally, the high prevalence and diversity of these parasites might be related to anthropogenic disturbance and activities in this area [53] or landscape characteristics in this area [54]. The prevalences of *Plasmodium* spp. and *Leucocytozoon* spp. between provinces were not significantly different, whereas *Trypanosoma* spp. in PHL was significantly high. This suggested that serious impacts caused by avian malaria and leucocytozoonosis should be a concern in NST, PHL and SUT. Although avian trypanosomiasis was harmless in domestic chickens, further investigation to maximize the information of these diseases should be considered. Since the molecular prevalence of *Trypanosoma* spp. was not high (33.33%), PHL might be the suitable area for sample collection.

This study described new genetic lineages of *Plasmodium* and *Leucocytozoon* (Table 2), indicating that the diversity of parasites in this area was high and even new species can be involved in the infections. Especially for *Leucocytozoon,* many of their lineages were found (Table 2), and some samples (AVC46, AVC56 and AVC58) might be undescribed *Leucocytozoon*. Co-infection and PCR failing to amplify the DNA of parasites were not new phenomena in avian malaria studies [55,56], and our findings reinforce the importance of combining microscopic and PCR-based techniques in the investigation of avian blood parasites. The sequencing of the PCR product helped for the identification of co-infection, whereas microscopic examination of thin blood smears provided the morphologic features for descriptions of the morphospecies. Therefore, further investigations are necessary to better understand whether we were dealing with a co-infection or if these finding can be due to the presence of a cryptic or a new *Leucocytozoon* species in backyard chickens in Thailand. It is noted that the detection of multiple detections can be difficult sometimes. This is because of the preferential amplification of *Plasmodium* co-infection with *Haemoproteus*. Additionally, co-infection between two strains at very different intensities can result in small double peaks [57].

Sixteen individuals were PCR-positive for *Plasmodium* sp. TSUB01. This lineage was first reported from the Eastern Slaty Thrush (*Turdus subalaris*, Turdidae, Passeriformes) in Brazil [8], which showed that *P. juxtanucleare* can spillover from domestic chicken to wild animals. Since *P. juxtanucleare* has a global distribution, when a suitable temperature is present, not only can the vectors develop but also the parasite can complete its life cycle and be transmitted to the next host, which can be domestic and wild birds [37,58,59]. Southern Thailand may be an excellent environment for the development and transmission of *P. juxtanucleare* and other vector-borne diseases between wild passerine birds and backyard chickens. The other seven individuals were PCR-positive to *P. gallinaceum* GALLUS01. This lineage was isolated from chickens (*Gallus gallus*, Phasianidae, Galliformes) in Vietnam more than 20 years ago [60]. This suggested that *P. gallinaceum* GALLUS01 can be transmitted among Southeastern Asian countries. Together with their high pathogenicity and high transmission rate [61], it was important to impose the preventive measures to minimize the occurrence of disease and prevent the loss of household incomes.

Three *Trypanosoma* species were reported to infect domestic chickens: *T. calmettei*, *T. gallinarum* and *T. numidae* [9]. The *Trypanosoma* sp. present in our samples had a short and slender trypomastigote (Figure 3G–I), which resembled *T. calmettei* [62]. The *Trypanosoma* sp. present in our samples can be readily differentiated from *T. gallinarum* and *T. numidae* by its smaller size (18.04 ± 1.55 µm), while *T. gallinaceum* had its length between 54.5 and 76.3 µm, and in *T. numidae* it is around 53 µm in length [9,63]. Due to the low number of trypomastigotes in the fresh blood smears, morphological identification was not possible. There were only two *Trypanosoma SSU* rRNA sequences isolated in the present study, and they were phylogenetically close to *T. avium* clade (Figure 6). However, these two sequences were different from the avian trypanosomes lineages described in the literature [27]. It is noteworthy that the previous article [27] described avian trypanosomes lineages based on the RAPD method. Thus, to define whether our sequences were new lineages, the RAPD analysis might be needed. The result of nested PCR (Table 1) indicates the existence of this parasite in Southern Thailand. Three trypomastigotes of trypanosomes found in buffy coat smears showed variation in their sizes and shapes (Figure 1F–H). Although this method was not recommended for describing morphospecies, this information may be evidence that domestic chickens can be infected by different species of *Trypanosoma* parasites. Therefore, the present study reinforces the importance of further studies addressing *Trypanosoma* parasites in the study area.

The rapid development of DNA-based molecular methods allowed us to understand the genetic diversity [64], population genetic structure [65] and parasite taxonomy [66]. The nested-PCR used in this study detected a higher number of *Plasmodium* spp., *Leucocytozoon* spp. and *Trypanosoma* spp. than the buffy-coat smear (Table 1), indicating low sensitivity of buffy-coat smear. However, the observation of Giemsa-stained buffy coat smears (Figure 1) was the evidence that this method can be used for the diagnosis of an infection of *Plasmodium* sp., *Leucocytozoon* sp. or *Trypanosoma* sp. Nevertheless, due to the initiation of the exflagellation process when mature *Leucocytozoon* gametocytes are in contact with air and the modifications that can be seen in the stained slides, this material is not recommended for descriptions of species—even though it can be used for avian blood parasite diagnosis [33]. On the other hand, the resources and facilities to perform molecular analysis might not be present and/or available in all veterinarian laboratories, and molecular-based diagnosis is more expensive than using microscopy. Therefore, even though molecular techniques were more specific and sensitive for avian haemosporidian diagnosis [32,33], the Giemsa-stained buffy-coat method can be recommended as a useful tool in the diagnosis of such infections.

The buffy coat smear was prepared by breaking the glass capillary tube right below the buffy coat layer, which results in a small portion of red blood cells being transferred to the glass slide [67]. This is a well-known methodology to investigate blood parasites, widely applied in the diagnosis of parasites of medical and veterinary importance [33,68,69,70], for both extra- and intracellular parasites, such as microfilaria, *Trypanosoma*, *Erlichia canis*, *Haemoproteus* and *Lankesterella*. However, different parasites might require some modification in the protocol to facilitate their diagnosis. For instance, human *Plasmodium* can be diagnosed with the application of the buffy coat method when blood samples are collected in capillary tubes containing orange acridine [71,72,73,74]. Previously, it was reported that the buffy coat method was not appropriate for the diagnosis of *Plasmodium* and *Leucocytozoon* (see [33] for detailed information). Here, we modified the proposed methodology that consisted of the preparation of smears and staining them with Giemsa. This improved the detection of these two parasites, making this methodology useful for the diagnosis of blood parasites in chickens. Additionally, this is a cheap and quick methodology, that can be easily applied in veterinary diagnosis.

## 5. Conclusions

This study aimed to investigate molecular prevalence and diversity of blood parasites infecting backyard chickens raised in Southern Thailand. This study showed a high molecular prevalence of *Plasmodium* spp. (64.91%) and *Leucocytozoon* spp. (89.47%). *Trypanossoma* parasites were also identified but in a lower prevalence (33.33%). Here we identified three new lineages of *Plasmodium* sp. (GALLUS47-49) and 13 new lineages of *Leucocytozoon* spp. (GALLUS50-62), confirming a high diversity of parasites infecting chickens in the study area. This prevalence and diversity might be due to the low biosecurity production system in which these chickens are raised. *Plasmodium gallinaceum* was identified in the few thin blood smears that were collected. These results highlight the importance of raising awareness to the livestock authorities and local farmers about the presence of these parasites, how they can compromise their production and the influence of poultry production on a bigger scale. Even though it is not recommended for detailed morphological analysis, buffy coat smears can be a useful tool for the diagnosis of avian blood parasites in veterinary clinics and laboratories, being cheap and easy to be performed. If more detailed information about parasite genetic diversity is necessary, then PCR-based methods should be employed. Further studies are necessary to better understand the pathogenicity and patterns of transmission of these parasites in backyard chickens. Additionally, potential vectors should be investigated in the area in order to elaborate and to test prophylactic methods to reduce the impact of infections.

## Figures and Tables

**Figure 1 animals-13-02798-f001:**
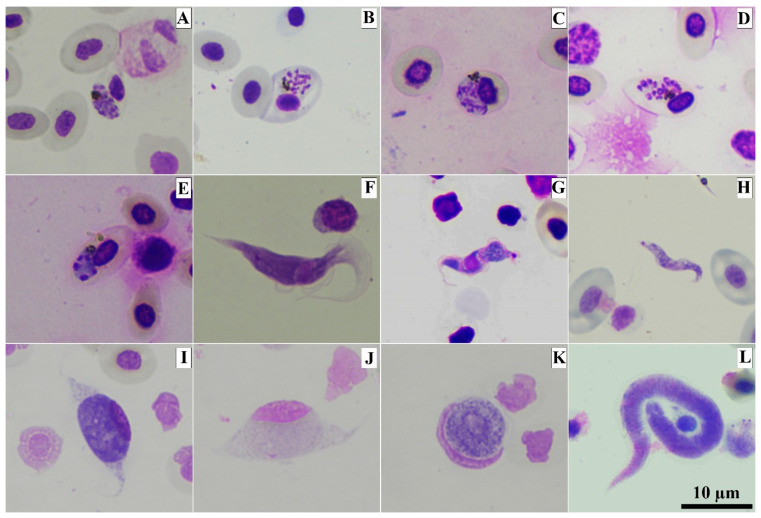
Photomicrographs of blood parasites present in buffy coat smears from domestic chickens (*Gallus gallus domesticus*). Erythrocytic meront of *Plasmodium gallinaceum* GALLUS01 (**A**–**E**), trypomastigote of avian *Trypanosoma* sp. (**F**–**H**) and gametocytes of *Leucocytozoon* sp.: elongate macrogametocyte (**I**), elongate microgametocyte (**J**) round macrogametocyte (**K**) and microfilaria of filarial nematodes (**L**). Methanol-fixed and stained with 10% Giemsa solution.

**Figure 2 animals-13-02798-f002:**
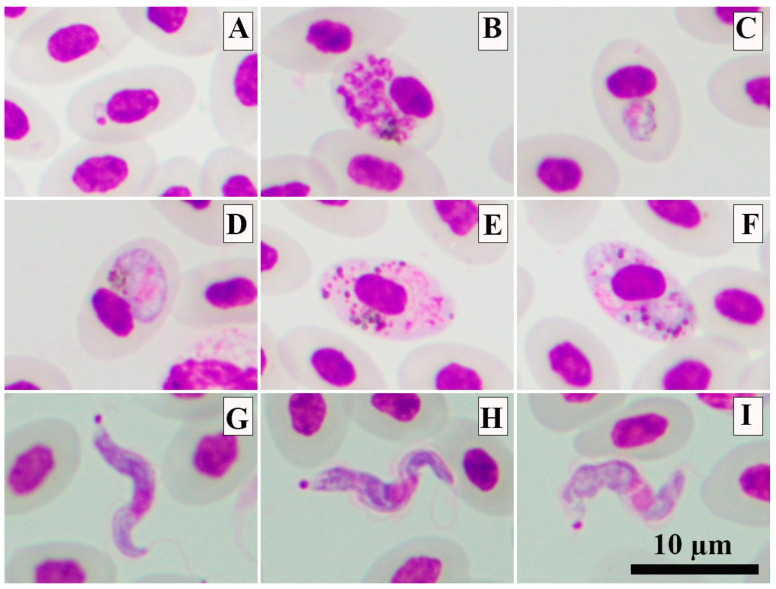
The photomicrographs of *Plasmodium gallinaceum* GALLUS01 (**A**–**E**) and *Trypanosoma* sp. (**G**–**I**) present in blood smears from domestic chickens. Trophozoite (**A**), mature erythrocytic meront (**B**)**,** growing gametocyte (**C**), mature microgametocyte (**D**) and mature macrogametocytes (**E**,**F**). Trypomastigotes of *Trypanosoma* sp. (**G**–**I**). Methanol-fixed and stained with 10% Giemsa solution.

**Figure 3 animals-13-02798-f003:**
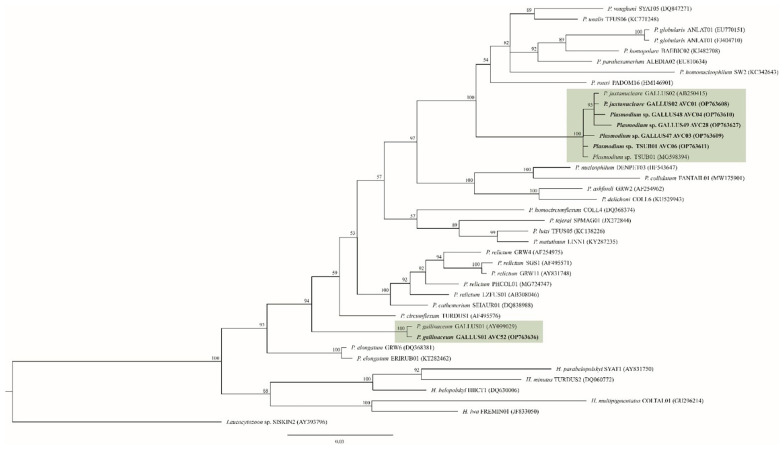
*Plasmodium* Bayesian phylogeny based on a fragment of cytochrome *b* gene (479 bp). The lineages isolated in this study are given in bold. Parasite lineages and isolate number are given after species names. GenBank accession numbers are given between brackets. Node values indicate percentages of posterior probabilities. *Plasmodium* lineages isolated from chickens are shown in the green boxes. *Leucocytozoon* sp. lineage SISKIN2 was an outgroup.

**Figure 4 animals-13-02798-f004:**
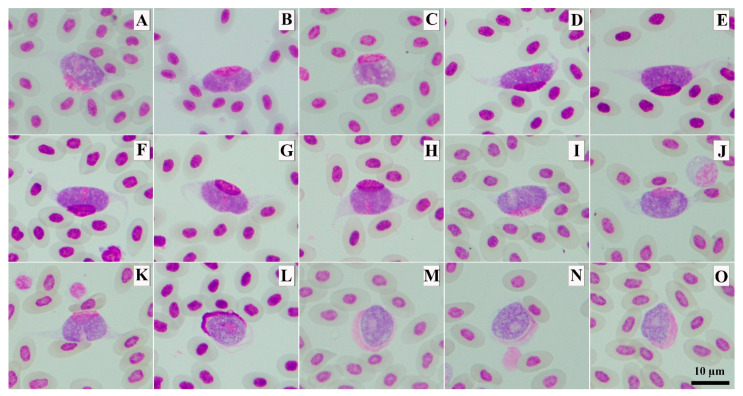
The photomicrographs of *Leucocytozoon* sp. macrogametocytes developed into fusiform host cells–parasite complexes (**A**–**K**) and roundish host cell–parasite complexes (**L**). Methanol-fixed and stained with 10% Giemsa solution. *Leucocytozoon* lineage GALLUS55 (**K**,**O**), GALLUS61 (**A**), GALLUS62 (**I**,**M**) and undescribed lineage (**B**–**H**,**J**,**L**,**N**).

**Figure 5 animals-13-02798-f005:**
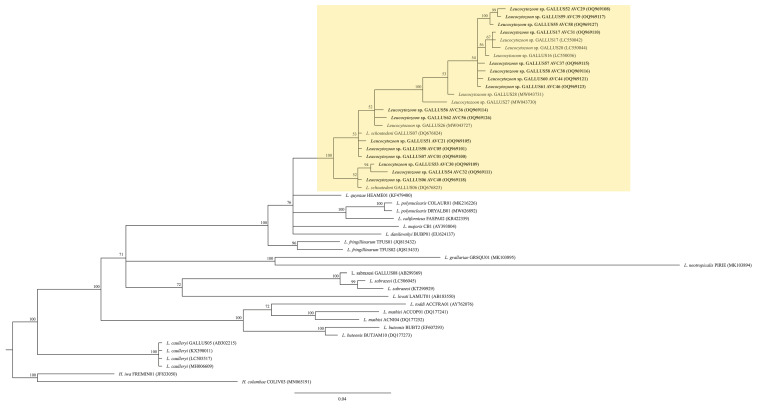
*Leucocytozoon* spp. bayesian phylogeny based on a fragment of cytochrome *b* gene (479 bp). The lineages isolated in this study are given in bold. Parasite lineages and isolation number are given after species names. GenBank accession numbers are given between brackets. Node values indicate percentages of posterior probabilities. *L. schouteni* clade is highlighted in yellow. *Haemoproteus iwa* lineage FREMIN01 and *H. columbae* COLIV03 were the outgroup.

**Figure 6 animals-13-02798-f006:**
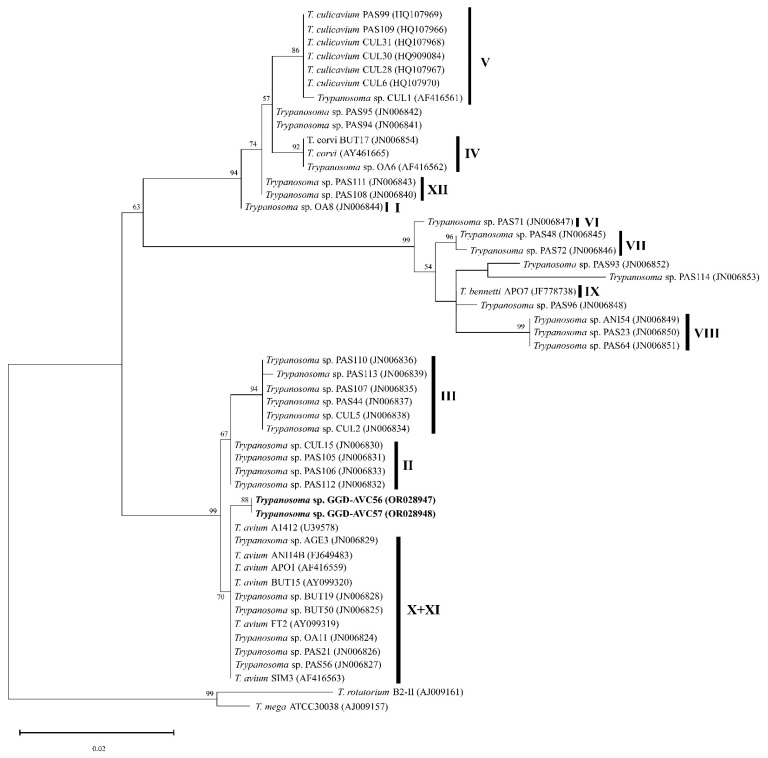
*Trypanosoma* spp. Maximun-likelihood phylogeny inference based on a fragment of the small subunit rRNA gene (798 bp). The sequences isolated in this study are given in bold. Vertical bars indicate the group of avian trypanosomes lineages (I–XII) following Zídková et al. (2012) [27]. Bootstrap values > 50% are shown at the branch points.

**Table 1 animals-13-02798-t001:** Number of blood parasites found in backyard chickens’ blood examined using buffy coat smear and nested-PCR.

	Buffy Coat Smear (n = 55)	Nested-PCR (n = 57)	Prevalence
*Plasmodium* sp.	5	37 ^a^	64.91%
*Leucocytozoon* sp.	48	51 ^b^	89.47%
*Trypanosoma* sp.	4	19 ^c^	33.33%
Microfilaria	15	NA	NA

NA = not available. ^a^ Two samples failed for nested-PCR of *Plasmodium* sp. (AVC40 & AVC49). ^b^ One sample failed for nested-PCR of *Leucocytozoon* sp. (AVC55). ^c^ Two samples failed for nested-PCR of *Trypanosoma* sp. (AVC07 & AVC29).

**Table 2 animals-13-02798-t002:** *Plasmodium* and *Leucocytozoon* lineages isolated from backyard chickens (*Gallus gallus domesticus*) raised in Southern Thailand, during June 2021–June 2022.

Linages	Isolates	Parasite Species	Host Breed	Locality		GenBank
				District	Province	
GALLUS01	AVC16	*Plasmodium gallinaceum*	N	LAS	NST	OP763632
	AVC32		N	KOR	PHL	OP763633
	AVC39		N	KOR	PHL	OP763634
	AVC46		N	THL	NST	OP763635
	AVC52		N	KHK	PHL	OP763636
	AVC56		N	KHK	PHL	OP763637
	AVC58		N	KHK	PHL	OP763638
GALLUS02	AVC01	*Plasmodium juxtanucleare*	H	THL	NST	OP763608
	AVC08		H	THL	NST	OP763613
	AVC09		H	THL	NST	OP763614
	AVC13		LH	THL	NST	OP763616
	AVC18		N	LAS	NST	OP763620
TSUB01	AVC06	*Plasmodium* sp.	H	THL	NST	OP763611
	AVC07		H	THL	NST	OP763612
	AVC11		LH	THL	NST	OP763615
	AVC14		LH	THL	NST	OP763617
	AVC15		LH	THL	NST	OP763618
	AVC17		N	LAS	NST	OP763619
	AVC20		N	LAS	NST	OP763621
	AVC22		FR	PHS	SUT	OP763622
	AVC23		FR	PHS	SUT	OP763623
	AVC24		FR	PHS	SUT	OP763624
	AVC25		FR	PHS	SUT	OP763625
	AVC26		FR	BND	SUT	OP763626
	AVC29		FR	BND	SUT	OP763628
	AVC31		N	KOR	PHL	OP763629
	AVC51		N	KHK	PHL	OP763630
	AVC57		N	KHK	PHL	OP763631
GALLUS47	AVC03	*Plasmodium* sp.	H	THL	NST	OP763609
GALLUS48	AVC04	*Plasmodium* sp.	H	THL	NST	OP763610
GALLUS49	AVC28	*Plasmodium* sp.	FR	BND	SUT	OP763627
GALLUS06	AVC40	*Leucocytozoon schoutedeni*	N	KOR	PHL	OQ969118
GALLUS07	AVC01	*Leucocytozoon schoutedeni*	H	THL	NST	OQ969100
	AVC07		H	THL	NST	OQ969102
	AVC22		FR	PHS	SUT	OQ969106
	AVC34		N	KOR	PHL	OQ969112
	AVC43		N	KOR	PHL	OQ969120
GALLUS17	AVC31	*Leucocytozoon* sp.	N	KOR	PHL	OQ969110
	AVC41		N	KOR	PHL	OQ969119
	AVC50		N	KHK	PHL	OQ969125
GALLUS50	AVC05	*Leucocytozoon* sp.	H	THL	NST	OQ969101
	AVC09		H	THL	NST	OQ969103
	AVC18		N	LAS	NST	OQ969104
GALLUS51	AVC21	*Leucocytozoon* sp.	FR	PHS	SUT	OQ969105
	AVC23		FR	PHS	SUT	OQ969107
GALLUS52	AVC29	*Leucocytozoon* sp.	FR	BND	SUT	OQ969108
	AVC45		N	KOR	PHL	OQ969122
GALLUS53	AVC30	*Leucocytozoon* sp.	FR	BND	SUT	OQ969109
GALLUS54	AVC32	*Leucocytozoon* sp.	N	KOR	PHL	OQ969111
GALLUS55	AVC35	*Leucocytozoon* sp.	N	KOR	PHL	OQ969113
	AVC58		N	KHK	PHL	OQ969127
GALLUS56	AVC36	*Leucocytozoon* sp.	N	KOR	PHL	OQ969114
GALLUS57	AVC37	*Leucocytozoon* sp.	N	KOR	PHL	OQ969115
GALLUS58	AVC38	*Leucocytozoon* sp.	N	KOR	PHL	OQ969116
GALLUS59	AVC39	*Leucocytozoon* sp.	N	KOR	PHL	OQ969117
GALLUS60	AVC44	*Leucocytozoon* sp.	N	KOR	PHL	OQ969121
GALLUS61	AVC46	*Leucocytozoon* sp.	N	THL	NST	OQ969123
GALLUS62	AVC56	*Leucocytozoon* sp.	N	KHK	PHL	OQ969126

Newly described lineages are given in bold. Chicken breed: N, native; H, hybrid; FR, fighting rooster; LH, laying hens. Locality: NST, Nakhon Si Thammarat; PHL, Phattalung; SUT, Surat Thani; LAS, Lan Saka; KOR, Kong Ra; THL, Thasala; KHK, Khun Khanun; PHS, Phrasaeng; BND, Ban Na Doem.

**Table 3 animals-13-02798-t003:** Morphometry of trypomastigotes of *Trypanosoma* sp. infected in backyard chickens (*Gallus gallus domesticus*, AVC49) raised in Southern Thailand.

Parameter *		*Trypanosoma* sp. (n = 5)			
		Mean	SD	Min	Max
AK	(µm^2^)	0.68	0.06	0.61	0.75
AN	(µm^2^)	4.45	0.79	3.62	5.63
AT	(µm^2^)	31.68	3.79	27.71	36.72
BW	(µm)	2.22	0.24	2.01	2.60
FF	(µm)	8.01	0.84	6.59	8.67
KN	(µm)	8.70	0.82	7.35	9.53
NA	(µm)	7.51	1.15	6.01	9.15
PA	(µm)	18.04	1.55	15.81	20.12
PK	(µm)	0.91	0.22	0.76	1.29
PN	(µm)	10.61	0.64	9.78	11.56

* AK = area of kinetoplast, AN = area of nucleus, AT = area of trypomastigote, BW = width of body through center of nucleus, FF = free flagellum, KN = center of nucleus to kinetoplast, NA = center of nucleus to anterior end, PA = total length without free flagellum, PK = posterior end to kinetoplast, PN = center of nucleus to posterior end.

## Data Availability

The data presented in this study are available in GenBank database (https://www.ncbi.nlm.nih.gov/genbank/, accessed on 23 May 2023) (accession numbers: OP763608-OP763638, OQ969100-OQ969127, OR028947-OR028948).

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
