# Peer review of "Prevalence and Diversity of Blood Parasites (Plasmodium, Leucocytozoon and Trypanosoma) in Backyard Chickens (Gallus gallus domesticus) Raised in Southern Thailand"

_animals, 2023, doi:10.3390/ani13172798_

Round 1
Reviewer 1 Report
Manuscript ID animals-2565117 entitled "Molecular detection of blood parasites (Plasmodium, Leucocytozoon, and Trypanosoma) in backyard chickens (Gallus gallus domesticus) raised in Southern Thailand"
COMMENTS:
The authors describe a study in which they investigated, by PCR-based testing and microscopic examination, the occurrence and diversity of blood parasites from three genera (Plasmodium, Leucocytozoon, and Trypanosoma) in backyard chickens (Gallus gallus domesticus) raised in three provinces located in Southern Thailand: Nakhon Si Thammarat, Phatthalung and Surat Thani. Research on the occurrence and community composition of avian hemoparasites is usually biased towards wild passerines, thus, this study adds information on new distribution areas and the diversity of parasite lineages for a poultry species, providing us with a better understanding of the community dynamics and potential transmission risk of vector-borne diseases in Southern Thailand, which could be of particular economic interest in the poultry sector. They found that the overall prevalence of Plasmodium, Leucocytozoon and Trypanosoma was 64.91%, 89.47% and 33.33%, respectively. The nested-PCR used in this study detected a higher prevalence of Plasmodium, Leucocytozoon and Trypanosoma than the buffy-coat smear, indicating a lower sensitivity of microscopic methods. In general, it is a well-written manuscript on a subject of general interest. However, I had the following general comments concerning the manuscript:
One of my main concerns in this study is the lack of clarity regarding the number of thin blood smears and buffy-coat smears performed. Only 11 out of 57 individuals (the sample size is also not too large for three study areas) had a thin blood smear and 55 out of 57 had a buffy-coat smear. Why is there this difference in the number and type of smears and the number of backyard chickens sampled? This sample size mismatch needs to be more clearly addressed throughout the manuscript. Furthermore, I do not understand why the authors rely on buffy-coat smears if they focus on the small fraction of white cells, when Plasmodium and some Leucocytozoon gametocytes parasitise red cells (erythrocytes), which are excluded from the analysis. This may underestimate prevalence when buffy-coat smears are examined microscopically. The more accurate option would be to analyse the prevalence and intensity of infection in non-centrifuged blood smears, where cells and parasites are evenly distributed.
I have detected some misused references, as they mention information that is not covered in the cited article even though this information is mentioned (cited) in that paper. For example, reference [10] and [23], lines 61 and 80 respectively. My job is not to check each reference one by one, please; authors have to make an ethical use of references when referring to previous work. They must check each reference and verify that the information is an original result in the corresponding cited article (except for bibliographic reviews).
The introduction could be more integrative, not making separate blocks for each genus of blood parasite.
The prevalence of blood parasites may vary depending on the seasonal variation (e.g. blood-sucking vectors may be more abundant in warmer months, Cosgrove et al., 2008 J. Anim. Ecol.) or on the sampling site (e.g. environments with more ponds or freshwater accumulations may generate more vector concentrations, Krama et al., 2015 J. Ornithol., Muriel et al., 2021 Diversity). As there are only 57 backyard chickens, it would be very useful if the authors could indicate more precisely the time of blood collection and the sample size in each study area. It would be very informative if the authors could compare the prevalence and diversity of genera and lineages between sampled areas, and if differences are detected, discuss possible reasons for this. On the other hand, the authors say that parasitemia was calculated as a percentage of the infected cells per 10,000 red blood cells. However, this would be unfeasible in buffy-coat smears as only white cells are present. Perhaps they are referring to the thin blood smears, but these are only 11 out of 57 individuals.
SPECIFIC COMMENTS:
Lines 13-23: Please check English and verb tenses. Furthermore, based on the journal's Instructions for Authors, the simple summary should contain a clear statement of the problem addressed, the aims and objectives, pertinent results, conclusions from the study and how they will be valuable to society.
Lines 2-4: If you use "molecular detection" in the title, it seems that you have excluded the microscopy part of your work. Perhaps you could use a more generic title referring to the prevalence and diversity of blood parasites.
Line 13: The term "important diseases" is vague. It should be more specific or highlight the problem more clearly.
Lines 13-14: The same applies to "important parasites". Moreover, why are Plasmodium and Leucocytozoon considered more important than other parasites?
Line 16: change “resulting in bigger economic impact” to “with a consequent higher economic impact”
Line 17: to report the molecular prevalence, lineage diversity and morphology of blood parasites.
Line 18: You must use the verb tenses well. What do you mean by high? You must specify how much. For some species, the prevalence may be high but for others it may be low.
Lines 19-20: Why were only the morphological characteristics of Plasmodium gallinaceum described?
Line 20: Tiny? Is it a unique characteristic of this species? Tiny is an unspecific adjective, how tiny compared to what?
Lines 21-23: Buffy coat is the portion of blood that contains concentrates of white blood cells, including monocytes and thrombocytes, so what is the point of checking these smears if Plasmodium and some Leucocytozoon are parasites that are usually found inside erythrocytes?
Lines 25-27: Try not to copy and paste literal sentences from the simple summary. You should elaborate more on this section. You should also take into account my previous comments on these lines.
Line 29: from the genus Plasmodium, Leucocytozoon and Trypanosoma
Line 30: microscopy and PCR-based detection methods
Lines 31-32: I hope to find a reason for this imbalance in the methods section. Even so, this sentence in the abstract provides little useful information about the work.
Lines 33-34: This sentence should be rewritten for clarity. In addition, “molecular investigation” is too generic a term.
Lines 34-36: Perhaps you should refer to method sensitivity. Have a look at this paper, I also recommend that it should be cited in the text (although not in the abstract):
· https://doi.org/10.1645/GE-2531.1
Lines 38-39: but there were only 11 thin blood smears, right? It is a bit confusing to know the real prevalence if you have only a few samples using different methodologies.
Line 40: change “host cell- parasite complexes” to “host cell-parasite complexes”
Line 43: Trypanosoma was found in 1 out of 11 thin blood smears, but there were only 11 thin blood smear from 57 backyard chickens. Therefore, this information is of little relevance. Maybe you can approach it differently.
Lines 43-45: it would be more informative to provide prevalence data.
Lines 45-47: If buffy coat smears are more focused on white cells, it does not make sense to look for erythrocyte (red cells) endoparasites such as Plasmodium and some cases of Leucocytozoon. From my point of view, parasite counts in buffy coat smears might be underestimating the prevalence and intensity of infection compared to molecular methods or thin blood smears.
Line 54: change “among 23 orders of host” to “among 23 host orders”
Line 56: change “globally” to “global”
Line 60; change “Plasmodium infection (known as avian malaria) in chickens” to “Plasmodium infection in chickens (known as avian malaria)”, as Plasmodium is not a genus that exclusively parasitises birds.
Line 61: this reference [10] focuses on the detection and molecular identification of Leucocytozoon and Plasmodium species, but never discusses symptoms related to malaria infection although they are mentioned on it.
Line 69: Leucocytozoonosis is not a disease endemic to Asia. This disease occurs worldwide, as you have indicated in Lines 52-53.
Line 70: See my comment for Line 61
Lines 78-80: The authors have merely transcribed the information given in this reference [23]. According to this reference, Valkiunas et al. 2011: “Avian trypanosomes are transmitted by wide variety of blood-sucking arthropods belonging to the Simuliidae, Culicidae,Ceratopogonidae, Hippoboscidae, and Dermanyssidae (Baker,1976; Molyneux, 1977; Miltgen and Landau, 1982; Voty ́pka andSvobodova ́ , 2004)”. The authors misuse the reference: “Avian trypanosomes are transmitted by blood-sucking arthropods belonging to families Simullidae, Culicidae, Ceratopogonidae, Hippoboscidae (Diptera) and Dermanyssidae (Mesostigmata) [23]”.
Lines 85-87: What about Trypanosoma?
Lines 91-92: This sentence has a result structure, so it could not be in this paragraph. In any case, it is a vague and weak sentence.
Lines 85-92: Predictions and hypotheses are usually presented in the last paragraph of the introduction. I would like to see more emphasis on the hypotheses, to make it more elaborated and clearer, and perhaps to elaborate a little on the possible difference between areas of study.
Lines 96-98: change to “From June 2021 to June 2022, a blood sample (maximum 1 mL) was taken from the brachial vein of 57 backyard chickens reared in Nakhon Si Thammarat (8°25'N, 99°58'E), Phatthalung (7°37'N, 100°4'E) and Surat Thani (9°7'N, 99°20'E)”.
Line 96: Anyway, I don't understand why you take such a high volume of blood, is there a reason?
Lines 96-98: More details about the timing of sampling and the study areas, as well as the sample size per area, would be necessary.
Line 101: change “where they were processed analysis” to “where the analyses were processed”
Lines 102-103: Why were thin blood smears taken from only 11 out of 57 individuals?
Lines 105-112: Indicate here the number of smears taken for each type.
Lines 114-115: 100 fields at 400x and another 100 at 1,000x magnification, or were they exactly the same fields?
Lines 115-116: The authors say that parasitemia was calculated as a percentage of the infected cells per 10,000 red blood cells. However, this would be unfeasible in buffy-coat smears as only white cells are present. Perhaps they are referring to the thin blood smears, but these are only 11 out of 57 individuals (see my comment above).
Lines 124-141: Why has Haemoproteus been excluded? This is a genus of blood parasite very common in chickens in Southeast Asia. See: https://doi.org/10.1016/j.actatropica.2020.105719. Please indicate in more detail the primers used. Why did you not use a Haemoproteus positive sample?
Line 154: change “lineages were” to “lineages were”
Lines 156-158 and 171-172: why did you use two outgroups?
Line 178: I would include an analysis to test the effect of the study area on the prevalence of different genera of blood parasites.
Line 182: You must indicate the version of the R programme
Lines 185-186: Rephrase this sentence (especially "had their blood investigated").
Lines 206-207: Why were these sequences excluded? It can be deduced from the electrophenogram. You are underestimating the diversity of lineages.
Lines 218-220: Low values are normal as you are underestimating the prevalence by looking for endoparasites in the wrong cells. Plasmodium is usually found inside the red blood cells that you have discarded.
Lines 223-224: microfilariae are not shown in figure 1K. That is a round macrogametocyte.
Line 248: Are you sure that the gametocytes shown in figures 2E and 2F are not Haemoproteus?
Line 317: Figure 6 needs to improve its sharpness and quality.
Line 322: You must specify that this occur in backyard chickens.
Line 336: species or lineages?
Lines 337-340: rephrase for clarity
Lines 340-342: you should consider and integrate this paper into your discussion: doi: 10.1017/s003118200500733x.
Line 343: delete one of the "However,"
Line 373: suggest?
Lines 377-378: it is not clear whether you are referring to the cited study or to your study.
Line 381: As I have commented on several occasions, you should be careful when referring to buffy-coat smears. This fraction of blood is not suitable for assessing the prevalence of erythrocyte-infecting parasites (e.g. Plasmodium).
Line 390-393: I consider this statement incorrect.
Lines 395-408: The conclusions should be reformulated based on my comments. They should also be more ambitious and highlight the strengths and novelties of this study.
Line 398: At several points in the manuscript, you refer to a "low biosecurity production system" in Thailand. However, I think you should consider other aspects when discussing your results. For example, it has been shown that in Myanmar (a country bordering Thailand) the prevalence and diversity of blood parasites may vary with anthropogenic disturbance: https://doi.org/10.3390/d13030111, or because of landscape characteristics in vector and host communities https://doi.org/10.1111/1365-2656.12805. Perhaps the authors should explore these ideas further when discussing their results. Are all sampling areas equal?
Line 410: A section on “Ethical Guidelines for the Use of Animals in Research” is missing. Licences? Reference numbers?
My comments on the Quality of the English Language are shown throughout the various corrections proposed in my review. It is true that the grammar and sentence structure should be improved in the abstracts and introduction, as well as in the first sections on material and methods.
Author Response
One of my main concerns in this study is the lack of clarity regarding the number of thin blood smears and buffy-coat smears performed. Only 11 out of 57 individuals (the sample size is also not too large for three study areas) had a thin blood smear and 55 out of 57 had a buffy-coat smear. Why is there this difference in the number and type of smears and the number of backyard chickens sampled? This sample size mismatch needs to be more clearly addressed throughout the manuscript. Furthermore, I do not understand why the authors rely on buffy-coat smears if they focus on the small fraction of white cells, when Plasmodium and some Leucocytozoon gametocytes parasitise red cells (erythrocytes), which are excluded from the analysis. This may underestimate prevalence when buffy-coat smears are examined microscopically. The more accurate option would be to analyse the prevalence and intensity of infection in non-centrifuged blood smears, where cells and parasites are evenly distributed.
We agree that it would be better to have more blood smears, but, unfortunately, it was not possible to prepare them from all the sampled individuals. This was mainly due to the field conditions in some places that resulted in blood clothing, which prevented the fresh blood smears preparation. However, we decided to present the results obtained anyway.
Why the authors rely on buffy-coat smears if they focus on the small fraction of white cells. This may underestimate prevalence when buffy-coat smears are examined microscopically. The more accurate option would be to analyze the prevalence and intensity of infection in non-centrifuged blood smears, where cells and parasites are evenly distributed.
Indeed, buffy-coat smears will focus on a small fraction of white cells. However, as has been shown by other authors that this is a reliable concentration method for blood parasites (see https://doi.org/10.1007/s00580-015-2161-5 and https://doi.org/10.1186/s13071-020-3984-8, among others).
That said, the buffy-coat method can increase the sensitivity and sensibility of blood parasite diagnoses. Additionally, this sample processing was used only to check infection status (infected or uninfected), and not to calculate the prevalence, which was calculated based on molecular results (lines 202-206). This is a standard method in avian blood parasite research and has been done by several authors in the last 20 years.
I have detected some misused references, as they mention information that is not covered in the cited article even though this information is mentioned (cited) in that paper. For example, reference [10] and [23], lines 61 and 80 respectively. My job is not to check each reference one by one, please; authors have to make an ethical use of references when referring to previous work. They must check each reference and verify that the information is an original result in the corresponding cited article (except for bibliographic reviews).
Thank you for pointing this out. We reviewed the references and included the missing ones.
The introduction could be more integrative, not making separate blocks for each genus of blood parasite.
The introduction was reviewed.
The prevalence of blood parasites may vary depending on the seasonal variation (e.g. blood-sucking vectors may be more abundant in warmer months, Cosgrove et al., 2008 J. Anim. Ecol.) or on the sampling site (e.g. environments with more ponds or freshwater accumulations may generate more vector concentrations, Krama et al., 2015 J. Ornithol., Muriel et al., 2021 Diversity).
We agree with that and appreciate that you pointed this out. The present study was the first time these areas were used for the investigation of blood parasites in backyard chickens. In order to understand the fluctuations of transmission in the study sites a detailed analysis of the surrounding areas and insect sampling and processing for these parasites would be necessary. Such a study can be conducted in the future and the obtained data can be discussed and compared with the findings presented here. In this study, we compare the prevalence between areas, using Fisher’s exact test (Line 190-193, 209-213).
As there are only 57 backyard chickens, it would be very useful if the authors could indicate more precisely the time of blood collection and the sample size in each study area.
The sample size was calculated previously to sampling for the three provinces together using ProMESA software Version 2.3.0.2, which showed it would be necessary to sample 42 individuals, and we sampled more than that.
Additionally, the sampling in each area was dependent on the availability of properties raising backyard chickens and how many could be sampled. The number of sampled animals in each locality was 23 in Nakhon Si Thammarat, 25 in Phatthalung and 9 in Surat Thani. This information was included in the manuscript (lines 203- 204).
Concerning the time of the year that the samples were collected on rainy season (June - September) and dry season (April - May) (lines: 105 - 106)
It would be very informative if the authors could compare the prevalence and diversity of genera and lineages between sampled areas, and if differences are detected, discuss possible reasons for this.
Thank you for the suggestion. The analysis was included. The methodology is explained in lines 190-193; results are presented in lines 209-213.
On the other hand, the authors say that parasitemia was calculated as a percentage of the infected cells per 10,000 red blood cells. However, this would be unfeasible in buffy-coat smears as only white cells are present. Perhaps they are referring to the thin blood smears, but these are only 11 out of 57 individuals.
You are right, the parasitemia was calculated only for the samples in which thin blood smears were available. We rephrased the sentence to clarify this issue (lines 125-127).
Lines 13-23: Please check English and verb tenses. Furthermore, based on the journal's Instructions for Authors, the simple summary should contain a clear statement of the problem addressed, the aims and objectives, pertinent results, conclusions from the study and how they will be valuable to society.
Thank you for your suggestion. The simple summary was revised.
Lines 2-4: If you use "molecular detection" in the title, it seems that you have excluded the microscopy part of your work. Perhaps you could use a more generic title referring to the prevalence and diversity of blood parasites.
Thank you for your suggestion, we appreciated it and changed the title.
Line 13: The term "important diseases" is vague. It should be more specific or highlight the problem more clearly.
The simple summary was reviewed.
Lines 13-14: The same applies to "important parasites". Moreover, why are Plasmodium and Leucocytozoon considered more important than other parasites?
There are several blood parasites that can be found in chickens (Plasmodium, Leucocytozoon, Haemoproteus, Trypanosoma and microfilaria). Some of them were better studied and it is well-known that they can be highly pathogenic to their avian hosts, which is the case of Plasmodium and Leucocytozoon. The simple abstract was modified to show this.
Line 16: change “resulting in bigger economic impact” to “with a consequent higher economic impact”
The simple summary was reviewed.
Line 17: to report the molecular prevalence, lineage diversity and morphology of blood parasites.
The simple summary was reviewed.
Line 16: You must use the verb tenses well. What do you mean by high? You must specify how much. For some species, the prevalence may be high but for others it may be low.
The simple summary was reviewed.
Lines 19-20: Why were only the morphological characteristics of Plasmodium gallinaceum described?
The simple summary was reviewed.
Line 20: Tiny? Is it a unique characteristic of this species? Tiny is an unspecific adjective, how tiny compared to what?
The simple summary was reviewed.
Lines 21-23: Buffy coat is the portion of blood that contains concentrates of white blood cells, including monocytes and thrombocytes, so what is the point of checking these smears if Plasmodium and some Leucocytozoon are parasites that are usually found inside erythrocytes?
As we mentioned before, the buffy coat is used to concentrate the parasites, which after centrifugation are located within white blood cells and the upper part of the red blood cells see https://doi.org/10.1007/s00580-015-2161-5 and https://doi.org/10.1186/s13071-020-3984-8, among others).
Lines 25-27: Try not to copy and paste literal sentences from the simple summary. You should elaborate more on this section. You should also take into account my previous comments on these lines.
The simple summary was reviewed.
Line 29: from the genus Plasmodium, Leucocytozoon and Trypanosoma
The sentence is revised.
Line 30: microscopy and PCR-based detection methods
The sentence is revised.
Lines 31-32: I hope to find a reason for this imbalance in the methods section. Even so, this sentence in the abstract provides little useful information about the work.
The questions regarding a low number of thin blood smears were previously addressed. And we respectfully disagree with this statement, since we are showing which methods were used to process collected samples.
Lines 33-34: This sentence should be rewritten for clarity. In addition, “molecular investigation” is too generic a term.
The sentence was reviewed.
Lines 34-36: Perhaps you should refer to method sensitivity. Have a look at this paper, I also recommend that it should be cited in the text (although not in the abstract): https://doi.org/10.1645/GE-2531.1
Thank you for your suggestion. The sensitivity of microscopy and PCR methods in the diagnosis of haemosporidian parasites is not a new topic and has been addressed by different authors. Some of them claimed that PCR-based methods are more reliable (https://doi.org/10.1645/GE-2531.1), while others state that the obtained results are comparable (http://dx.doi.org/10.1645/GE-1570.1).
Additionally, when a standard microscopy method was compared to buffy-coat techniques, the latter showed to have a higher sensibility and sensitivity for some parasites than others (https://doi.org/10.1186/s13071-020-3984-8). In this study, we slightly modified the methodology for the buffy-coat method in an attempt to increase its sensibility and sensitivity, but calculating such parameters was not the goal of this study. Maybe this is something that can be addressed in future studies, and this modified method can also be compared with the thin blood smears results.
Lines 38-39: but there were only 11 thin blood smears, right? It is a bit confusing to know the real prevalence if you have only a few samples using different methodologies.
We report the prevalence based on nested PCR results.
Line 40: change “host cell- parasite complexes” to “host cell-parasite complexes”
Corrected.
Line 43: Trypanosoma was found in 1 out of 11 thin blood smears, but there were only 11 thin blood smears from 57 backyard chickens. Therefore, this information is of little relevance. Maybe you can approach it differently.
Of these 11 thin blood smears, one shows trypomastigotes of Trypanosoma. We included the prevalence obtained by PCR (line 45).
Lines 43-45: it would be more informative to provide prevalence data.
The information was included.
Lines 45-47: If buffy coat smears are more focused on white cells, it does not make sense to look for erythrocyte (red cells) endoparasites such as Plasmodium and some cases of Leucocytozoon. From my point of view, parasite counts in buffy coat smears might be underestimating the prevalence and intensity of infection compared to molecular methods or thin blood smears.
This question was previously addressed.
Line 54: change “among 23 orders of host” to “among 23 host orders”
Corrected.
Line 56: change “globally” to “global”
Corrected.
Line 60; change “Plasmodium infection (known as avian malaria) in chickens” to “Plasmodium infection in chickens (known as avian malaria)”, as Plasmodium is not a genus that exclusively parasitises birds.
Corrected.
Line 61: this reference [10] focuses on the detection and molecular identification of Leucocytozoon and Plasmodium species, but never discusses symptoms related to malaria infection although they are mentioned on it.
Thank you for pointing this out, we reviewed the reference and updated it.
Line 69: Leucocytozoonosis is not a disease endemic to Asia. This disease occurs worldwide, as you have indicated in Lines 52-53.
The sentence “and is an endemic disease in south-eastern Asian countries” is deleted.
Line 70: See my comment for Line 61
We reviewed the reference and updated it Reference numbers 24 and 25 were included.
Lines 78-80: The authors have merely transcribed the information given in this reference [23]. According to this reference, Valkiunas et al. 2011: “Avian trypanosomes are transmitted by wide variety of blood-sucking arthropods belonging to the Simuliidae, Culicidae,Ceratopogonidae, Hippoboscidae, and Dermanyssidae (Baker,1976; Molyneux, 1977; Miltgen and Landau, 1982; Voty ́pka andSvobodova ́ , 2004)”. The authors misuse the reference: “Avian trypanosomes are transmitted by blood-sucking arthropods belonging to families Simullidae, Culicidae, Ceratopogonidae, Hippoboscidae (Diptera) and Dermanyssidae (Mesostigmata) [23]”.
Thank you for pointing this out. We included the missing information (Ref 13-16).
Lines 85-87: What about Trypanosoma?
Thank you. Sentence is deleted.
Lines 91-92: This sentence has a result structure, so it could not be in this paragraph. In any case, it is a vague and weak sentence.
The paragraph is reviewed.
Lines 85-92: Predictions and hypotheses are usually presented in the last paragraph of the introduction. I would like to see more emphasis on the hypotheses, to make it more elaborated and clearer, and perhaps to elaborate a little on the possible difference between areas of study.
The paragraph is reviewed.
Lines 96-98: change to “From June 2021 to June 2022, a blood sample (maximum 1 mL) was taken from the brachial vein of 57 backyard chickens reared in Nakhon Si Thammarat (8°25'N, 99°58'E), Phatthalung (7°37'N, 100°4'E) and Surat Thani (9°7'N, 99°20'E)”.
Corrected.
Line 96: Anyway, I don't understand why you take such a high volume of blood, is there a reason?
From bigger birds it is easier to collect blood using syringe and needle, than to simple punch the vein and collected a few drops (as usually is done for passerines, specially the small ones). It is safe to withdraw this volume of blood, which can reach up to 1% of body weight of the birds (https://doi.org/10.1053/j.jepm.2010.01.006).
Lines 96-98: More details about the timing of sampling and the study areas, as well as the sample size per area, would be necessary.
Thank you for the suggestion. The information was included (lines 105 and 106, and 200-202).
Line 101: change “where they were processed analysis” to “where the analyses were processed”
Corrected.
Lines 102-103: Why were thin blood smears taken from only 11 out of 57 individuals?
This question was previously addressed.
Lines 105-112: Indicate here the number of smears taken for each type.
The information is added (line 113-114).
Lines 114-115: 100 fields at 400x and another 100 at 1,000x magnification, or were they exactly the same fields?
Using 400x magnification allows the quickly identification of some parasites, such as Leucocytozoon, Trypanosoma and microfilaria. This is mainly because they are bigger, and can be concentrated in certain parts of the thin blood smears (Leucocytozoon and microfilaria tend to be in the edge). However, this does not allow parasitemia calculation or the identification of small parasites, such as young stages of Plasmodium, making necessary to also perform microscopical analysis at 1000x magnification.
It is also important to mention that this protocol is considered to be the gold standard in avian malaria. The sentence was reviewed to clarify this matter.
Lines 115-116: The authors say that parasitemia was calculated as a percentage of the infected cells per 10,000 red blood cells. However, this would be unfeasible in buffy-coat smears as only white cells are present. Perhaps they are referring to the thin blood smears, but these are only 11 out of 57 individuals (see my comment above).
We performed this calculation on thin blood smears available.
Lines 124-141: Why has Haemoproteus been excluded? This is a genus of blood parasite very common in chickens in Southeast Asia. See: https://doi.org/10.1016/j.actatropica.2020.105719. Please indicate in more detail the primers used. Why did you not use a Haemoproteus positive sample?
Thank you for raising this question. Haemoproteus parasites were not excluded from our analysis. We used Bensch et al., 2000 and Hellgren et al., 2004 protocols to amplify haemosporidian DNA. In this protocol, the first reaction amplifies the DNA from Plasmodium, Haemoproteus and Leucocytozoon, and the nested reaction different primers are used to amplify Plasmodium/Haemoproteus, and Leucocytozoon.
All samples with a positive amplification were sequenced, and no Haemoproteus DNA was detected, as well as the presence of double peaks in the electropherograms (which would indicate a co-infection).
Additionally, we did not detect the presence of Haemoproteus parasites in the buffy coat smears and in the thin blood smears.
Line 154: change “lineages were” to “lineages were”
Corrected.
Lines 156-158 and 171-172: why did you use two outgroups?
For Leucocytozoon phylogenetics these two outgroups relate to each other. For Trypanosoma phylogenetics, we follow the previous report (https://doi,org. 10.1016/j.meegid.2011.10.022).
Line 178: I would include an analysis to test the effect of the study area on the prevalence of different genera of blood parasites.
We compare prevalence of parasites between area by using Fisher’s exact test. The methodology (lines 190-193) and the results (lines 209-213) were included.
Line 182: You must indicate the version of the R programme
Included.
Lines 185-186: Rephrase this sentence (especially "had their blood investigated").
Corrected.
Lines 206-207: Why were these sequences excluded? It can be deduced from the electrophenogram. You are underestimating the diversity of lineages.
They were excluded because it was not possible to deduct them from the electropherogram.
Lines 218-220: Low values are normal as you are underestimating the prevalence by looking for endoparasites in the wrong cells. Plasmodium is usually found inside the red blood cells that you have discarded.
The application of the buffy coat method consists of concentrating parasites in one layer of cells. After centrifugation the parasites will be in the top of the red blood cells, within the leukocytes layer and right above, in the plasma. When the capillary tube is broken, the top layer of red blood cells will also be transferred to the glass slides (as indicate by https://doi.org/10.1007/s00580-015-2161-5 and https://doi.org/10.1186/s13071-020-3984-8, among others).
It also worth mentioning, that similar protocol is used for the diagnosis of human malaria (i.e., https://www.sciencedirect.com/science/article/abs/pii/S1383576907000943). Of course, different techniques will have different sensitivity and specificity, but we do believe that using buffy coat smears is an easy, quick and cheap parasitological diagnosis tool, that can be applied in the daily routine of veterinary clinics and laboratories.
Lines 223-224: microfilariae are not shown in figure 1K. That is a round macrogametocyte.
Thank you for pointing this out. The image was corrected.
Line 248: Are you sure that the gametocytes shown in figures 2E and 2F are not Haemoproteus?
We confirmed it is not Haemoproteus by using multiplex-PCR (https://doi.org/10.1007/s00436-018-6153-7). Additionally, our finding is similar to the previous report (the picture from graphical abstract) https://doi.org/10.1016/j.vetpar.2017.05.002.
Line 317: Figure 6 needs to improve its sharpness and quality.
We improve Figure 6 (resolution 1,000 ppi).
Line 322: You must specify that this occur in backyard chickens.
Corrected.
Line 336: species or lineages?
Lineages (line 350)
Lines 337-340: rephrase for clarity
The sentence is revised (line 352-354).
Lines 340-342: you should consider and integrate this paper into your discussion: doi: 10.1017/s003118200500733x.
The article is integrated in discussion (line 361-364).
Line 343: delete one of the "However,"
Corrected.
Line 373: suggest?
We used “indicates” (line 391).
Lines 377-378: it is not clear whether you are referring to the cited study or to your study.
This is our suggestion. The sentence was rephrased (lines 395-397).
Line 381: As I have commented on several occasions, you should be careful when referring to buffy-coat smears. This fraction of blood is not suitable for assessing the prevalence of erythrocyte-infecting parasites (e.g. Plasmodium).
We followed your suggestion and added a paragraph discussing the pros and cons of buffy coat diagnosis (lines 414-428).
Line 390-393: I consider this statement incorrect.
Thank you for your suggestion, but we respectfully disagree. As in any other diagnosis methodology, this one also has its pros and cons (which were addressed as previously suggested). When choosing a diagnosis methodology, the professional should be aware of its downsides and use complementary methods that would be more adequate.
For instance, the buffy coat method (without the smear) is not recommended for Leucocytozoon diagnosis (as indicated by https://doi.org/10.1186/s13071-020-3984-8). This is mainly because the exflagellation of Leucocytozoon parasites does not always occur and the parasites will look like leukocytes. However, when the buffy coat smears is stained, the parasites can be seen and one of the downsides can be addressed.
As has been extensively addressed in several avian malaria studies, the most reliable form to diagnosis blood parasites in birds is combining different methodologies, as we are suggesting in the present study.
Lines 395-408: The conclusions should be reformulated based on my comments. They should also be more ambitious and highlight the strengths and novelties of this study.
Conclusion was reviewed.
Line 398: At several points in the manuscript, you refer to a "low biosecurity production system" in Thailand. However, I think you should consider other aspects when discussing your results. For example, it has been shown that in Myanmar (a country bordering Thailand) the prevalence and diversity of blood parasites may vary with anthropogenic disturbance: https://doi.org/10.3390/d13030111, or because of landscape characteristics in vector and host communities https://doi.org/10.1111/1365-2656.12805. Perhaps the authors should explore these ideas further when discussing their results. Are all sampling areas equal?
The suggestion was addressed, and a discussion was included in lines 340-342.
Line 410: A section on “Ethical Guidelines for the Use of Animals in Research” is missing. Licences? Reference numbers?
The information was included.
Reviewer 2 Report
Overall, this is an interesting study about the importance of Plasmodium, Leucocytozoon and Trypanosoma in backyard chickens prevalence. The high prevalence of Plasmodium spp. (64.91%) revealed and Leucocytozoon spp. (89.47%) in backyard chickens by this study is remarkably alarming. My comments below indicate things that need to be revised or explained.
1. Line 204, six Plasmodium and 24 Leucocytozoon sequences showed double peak in the electropherogram being considered as co-infections. How do you rule out the double peaks is not caused by the poor specificity of PCR?
2. Line 220, the PCR-protocol used in the study failed to amplify a few microscopically positive samples…… Microscopically positive samples is the gold standard for pathogen detection. Nested PCR is often more sensitive, but the results show that some microscopically positive samples can not be detected by nested PCR. How do you explain that?
3. Table 3, suggested that the table should be displayed in a diagram, which would be easier to understand. Such as box and whiskers with asterisks.
4. The results showed that the prevalence rate of blood parasites was high in the collected backyard chickens. Have you ever been treated with drugs? Does the blood parasite have drug resistance?
English writing is fine. I don't have any better suggestions.
Author Response
Overall, this is an interesting study about the importance of Plasmodium, Leucocytozoon and Trypanosoma in backyard chickens prevalence. The high prevalence of Plasmodium spp. (64.91%) revealed and Leucocytozoon spp. (89.47%) in backyard chickens by this study is remarkably alarming. My comments below indicate things that need to be revised or explained.
- Line 204, six Plasmodium and 24 Leucocytozoon sequences showed double peak in the electropherogram being considered as co-infections. How do you rule out the double peaks is not caused by the poor specificity of PCR?
These primers have been used in chickens before (https://doi.org/10.34044/j.anres.2020.54.6.04, 10.2478/jvetres-2022-0049)
- Line 220, the PCR-protocol used in the study failed to amplify a few microscopically positive samples…… Microscopically positive samples is the gold standard for pathogen detection. Nested PCR is often more sensitive, but the results show that some microscopically positive samples can not be detected by nested PCR. How do you explain that?
This may be related to low parasitemia in PCR-failed samples.
- Table 3, suggested that the table should be displayed in a diagram, which would be easier to understand. Such as box and whiskers with asterisks.
We would like to present the information in Table.
- The results showed that the prevalence rate of blood parasites was high in the collected backyard chickens. Have you ever been treated with drugs? Does the blood parasite have drug resistance?
These chickens have never been treated. For the information of drug resistance, we did not see any report.
Reviewer 3 Report
This study aims to report the molecular prevalence, parasite lineages and morphology of parasites. While going through the manuscript, i have following suggestions
Question 1. Sample Size and Representativeness:
The abstract mentions that only 57 backyard chickens were sampled. Given the potential variability in disease prevalence, isn't this sample size too small to draw meaningful conclusions about the prevalence of blood parasites in backyard chickens across three provinces? How were the 57 backyard chickens selected for sampling? Was there a random or systematic sampling approach?
Question 2. Sensitivity and Specificity of Diagnostic Tools:
Sensitivity and specificity of the nested-PCR and buffy coat smear methods was not discussed. False positives and false negatives can significantly impact the accuracy of disease prevalence estimates. Please mention if there is no false positive outcome.
Question 3. The study identified several genetic lineages of Plasmodium, Leucocytozoon, and Trypanosoma. Could these genetic variants differ in terms of pathogenicity, transmission, or host range? The study did not investigate potential interactions or synergistic effects between different blood parasites.
Question 4 . Economic Impact Assessment:
The abstract highlights the economic impact of infectious diseases on industrial poultry production. How these estimates were derived, and were they validated against real-world data?
Question 5. Include future recommendations for improving biosecurity measures and disease control strategies in backyard chicken production systems? Highlight any gaps in the current understanding of these parasites, both in terms of their impact on domestic chickens and their potential interactions.
Question 6. This study does not include any clinical manifestations and potential health consequences of co-infections in humans as well. Reconsider please.
Question 7. The methodology section of this study did mention any ethical approval statement. Were ethical considerations regarding animal welfare/handling/sampling and biosecurity taken?
Question 8. The methodology mentions using 1,000 replicates of bootstrap for phylogenetic analysis. Is this number sufficient to ensure robustness in the results?
Question 9. Future Directions: What are the potential follow-up studies that could build upon these findings? Are there specific aspects of the parasites' life cycles, host interactions, or transmission dynamics that warrant further investigation?
Question 10. The study seems to be conducted over a one-year period. How might seasonal variations in parasite prevalence affect the results?
Some minor revisions required
Author Response
Question 1. Sample Size and Representativeness:
The abstract mentions that only 57 backyard chickens were sampled. Given the potential variability in disease prevalence, isn't this sample size too small to draw meaningful conclusions about the prevalence of blood parasites in backyard chickens across three provinces? How were the 57 backyard chickens selected for sampling? Was there a random or systematic sampling approach?
The sample size was calculated previously to sampling for the three provinces together using ProMESA software Version 2.3.0.2, which showed it would be necessary to sample 42 individuals, and we sampled more than that.
This was a random sampling, since it was dependent on the availability of properties raising backyard chickens and how many could be sampled. The number of sampled animals in each locality was included in the manuscript (lines 203-205).
Question 2. Sensitivity and Specificity of Diagnostic Tools:
Sensitivity and specificity of the nested-PCR and buffy coat smear methods was not discussed. False positives and false negatives can significantly impact the accuracy of disease prevalence estimates. Please mention if there is no false positive outcome.
We apply the nested-PCR method from previous report. This method is generally used. We use non-template control for PCR test. Thus, there is no false positive results in this study. Additionally, number of parasites found in buffy coat smear is the evidence of parasite infection.
Question 3. The study identified several genetic lineages of Plasmodium, Leucocytozoon, and Trypanosoma. Could these genetic variants differ in terms of pathogenicity, transmission, or host range? The study did not investigate potential interactions or synergistic effects between different blood parasites.
Thank you very much for your comment. Indeed, this would be a very interesting study to be conducted in the future, especially because it would really indicate what are the main parasites that could influence poultry production.
For this type of study it would be necessary to collect other information from the sampled animals, such as body weight, hematocrit, among others, as well as to clinically examinate each individual. This was not done in the present study, but we will consider this in future ones.
Question 4. Economic Impact Assessment:
The abstract highlights the economic impact of infectious diseases on industrial poultry production. How these estimates were derived, and were they validated against real-world data?
It is known that some Plasmodium and Leucocytozoon species can cause death in infected individuals, emaciated chickens that result in low meat production, and decreased egg production. The impact in backyard chickens may be not that high. However, the presence of parasites and competent vectors in the area, increases the chances of these pathogens spreading to the poultry production, which represents a risk.
Question 5. Include future recommendations for improving biosecurity measures and disease control strategies in backyard chicken production systems? Highlight any gaps in the current understanding of these parasites, both in terms of their impact on domestic chickens and their potential interactions.
The backyard production systems is the low-cost production system. It is difficult to change the farmer behaviors. Additionally, it is impossible to suddenly change the biosecurity level. To control these diseases, chicken feeds mixing with anti-parasitic herb may be the alternative way.
Question 6. This study does not include any clinical manifestations and potential health consequences of co-infections in humans as well. Reconsider please.
We did not assess the clinical sign of chickens, but none of the studied animals presented obvious manifestations of diseases. Unfortunately, it will not be possible to address this in the present study, but for sure it will be considered in future studies.
It worth mentioning that these parasites are exclusively found in chickens, and do not represent any harm to humans.
Question 7. The methodology section of this study did mention any ethical approval statement. Were ethical considerations regarding animal welfare/handling/sampling and biosecurity taken?
The information was included in the proper section.
Question 8. The methodology mentions using 1,000 replicates of bootstrap for phylogenetic analysis. Is this number sufficient to ensure robustness in the results?
1,000 replication of bootstrap is acceptable. Normally, we accept at 250 replications.
Question 9. Future Directions: What are the potential follow-up studies that could build upon these findings? Are there specific aspects of the parasites' life cycles, host interactions, or transmission dynamics that warrant further investigation?
This was the first time that this area was studied in Thailand, and now we can have at least one picture of what happens in the study area: high diversity and prevalence of parasites. For further studies, several issues can be addressed, such as: (i) estimation of economic loses caused by these parasites; (ii) potential vectors involved in their transmission; (iii) pathogenicity of avian blood parasites; (iv) investigation of suitable mosquito repellent plants that can be used to prevent insects to bit infected individuals; (v) effects of co-infections in the healthy of these animals; among others.
Question 10. The study seems to be conducted over a one-year period. How might seasonal variations in parasite prevalence affect the results?
In Southern Thailand, rainy season is longer than dry season. In this study, we collect more samples in the rainy season than in the dry season. Thus, we did not compare the number of parasite infection between rainy and dry season because it would be necessary to design a study focusing on this, with a systematic sampling method. However, this is the interesting topic, that can be further investigated.
Reviewer 4 Report
The authors present an interesting study on blood parasites of backyard chickens raised in Southern Thailand. The bright side of the manuscript is to provide the blood parasites of the species in Southern Thailand and is likely to contribute to the healthy production of backyard chickens. However, some points are missing (mentioned below) in the manuscript and some parts of the manuscript are not easy to understand. Because of these reasons, major concerns are raised. Therefore, I would like to make some suggestions to improve the quality of the paper as below:
General Comments
Some parts of the manuscript are not easy to understand (mentioned below in specific comments). There are many long sentences and wordiness. This situation disrupts the flow of the subject and the continuity of the reading. Because of this reason, authors should re-reconsider writing some parts of the manuscript.
The Introduction section needs structural changes. The Discussion section should be enriched with a more theoretical interpretation and relate the present results with additional concepts. For instance, the study results can be discussed in the framework of blood parasite diversity in studies related to poultry and free-ranging bird species from different countries in the broader context. In this context, the discussion section also needs structural changes. Moreover, the limitations of the study should be given in the conclusion section.
Specific Comments
Simple Summary
Lines 13-14: Please merge these two sentences “Blood parasites causes important diseases in chickens. The important parasites are hemosporidian parasites, including Plasmodium spp. and Leucocytozoon spp.”
Abstract
Lines 25-47: I think the abstract needs to be rephrased and improved. In my opinion, it is good to start with the problem examined in the study. Within this context, the main problem that is examined by the authors should be explained in 1-2 sentences at the beginning of the abstract. In this context, this sentence “In, backyard production, chickens are raised in a low biosecurity system, on which infectious diseases can occur and may shed to the industrial poultry production, leading to a bigger economic impact” mentions the main problem. Then, the methods and the main results should be given briefly. This can be followed by the main findings of the study. Finally, what is the importance of the results and how the results contribute to further studies should be written down. In my opinion, it is always good to finish the abstract with such a sentence. Furthermore, authors may also say in 1-2 sentences that their findings contribute to further studies and healthy backyard production. In this way, the bridge between the problem and the solution found by the authors would be stronger.
Introduction
Line 56: “with a globally distribution” -> “, has a global distribution”.
Lines 85-92: This part of the paper is important since the authors should explain the purpose of the study and their hypothesis (I mean; what is the problem and what did you do to solve this problem) are given here. The authors explained the purpose of the study, but I think, this part of the paper should be rephrased. Please start with a brief description of the problem, in this context low biosecurity system of the backyard production. This sentence should be followed with the importance of the molecular detection of blood parasites and supporting the results with microscopic examination. After, the purpose of the study should be described. For instance, “Therefore, in this study, we aimed to examine the presence of Plasmodium, Leucocytozoon and Trypanosoma in backyard chickens raised in three provinces in Southern Thailand (Nakhon Si Thammarat, Phatthalung and Surat Thani). In this way, the bridge between the problem and the study aim would be stronger.
Materials and Methods
Line 113: Microscopical -> Microscopic
Line 137: follow -> followed
Line 154: were -> were
Line 182: R programming -> R
Please add the related references for each method and software used for the study.
Results
Lines 185-187: “During June 2021 and June 2022, 57 backyard chickens (Gallus gallus domesticus) had their blood investigated for the presence of Plasmodium, Leucocytozoon and Trypanosoma infections.” - > A total of 57 backyard chickens’ (Gallus gallus domesticus) blood samples were investigated to detect the presence of Plasmodium, Leucocytozoon and Trypanosoma infections.”
Lines 187-189: “This included native chickens (n = 33), hybrid chickens (n = 10), laying hens (n = 5) and fighting roosters (n = 9).” Did you find any difference among these groups?
Line 218: Microscopical -> Microscopic
Discussion
Lines 322-324: “This study showed a high molecular prevalence of avian malaria (P. gallinaceum and P. juxtanucelare) and leucocytozoonosis (Leucocytozoon spp.) in Nakhon Si Thammarat, Phatthalung and Surat Thani provinces in the Southern of Thailand”. Please rephrase this sentence.
Line 308: I think, this paragraph should start with the importance of molecular methods. Such as sentence can be added at the beginning of the paragraph “ The rapid development of DNA-based molecular methods allows us to understand the species' taxonomy, genetic diversity, population genetic structure, conservation genetics, and parasites”. Also, these references can be added here (doi: doi.org/10.3390/pathogens10020103, 10.3390/biology12030401, 10.3390/ani10081441, 10.3390/pathogens12050712). After that “In this context, the nested-PCR used detected a higher number of Plasmodium spp., Leucocytozoon spp. and Trypanosoma spp. than the buffy-coat smear…”
Conclusion
The conclusion should be rephrased as follows; please start with a brief description of the study (the aim of the study with a sentence), explain the main findings of the study briefly (the results that the authors found), explain how your results contribute to healthy backyard production etc. with 2-3 sentences, explain the limitations of the study and describe the future remarks briefly.
Some parts of the manuscript are not easy to understand (mentioned below in specific comments). There are many long sentences and wordiness. This situation disrupts the flow of the subject and the continuity of the reading.
Author Response
General Comments
Some parts of the manuscript are not easy to understand (mentioned below in specific comments). There are many long sentences and wordiness. This situation disrupts the flow of the subject and the continuity of the reading. Because of this reason, authors should re-reconsider writing some parts of the manuscript.
Thank you for your suggestion. We did our best to make the flow of the text better.
The Introduction section needs structural changes. The Discussion section should be enriched with a more theoretical interpretation and relate the present results with additional concepts. For instance, the study results can be discussed in the framework of blood parasite diversity in studies related to poultry and free-ranging bird species from different countries in the broader context. In this context, the discussion section also needs structural changes. Moreover, the limitations of the study should be given in the conclusion section.
Thank you for the suggestion, we did out best to improve the manuscript.
Specific Comments
Simple Summary
Lines 13-14: Please merge these two sentences “Blood parasites causes important diseases in chickens. The important parasites are hemosporidian parasites, including Plasmodium spp. and Leucocytozoon spp.”
The simple summary was reviewed.
Abstract
Lines 25-47: I think the abstract needs to be rephrased and improved. In my opinion, it is good to start with the problem examined in the study. Within this context, the main problem that is examined by the authors should be explained in 1-2 sentences at the beginning of the abstract. In this context, this sentence “In, backyard production, chickens are raised in a low biosecurity system, on which infectious diseases can occur and may shed to the industrial poultry production, leading to a bigger economic impact” mentions the main problem. Then, the methods and the main results should be given briefly. This can be followed by the main findings of the study. Finally, what is the importance of the results and how the results contribute to further studies should be written down. In my opinion, it is always good to finish the abstract with such a sentence. Furthermore, authors may also say in 1-2 sentences that their findings contribute to further studies and healthy backyard production. In this way, the bridge between the problem and the solution found by the authors would be stronger.
The abstract was reviewed.
Introduction
Line 56: “with a globally distribution” -> “, has a global distribution”.
Corrected.
Lines 85-92: This part of the paper is important since the authors should explain the purpose of the study and their hypothesis (I mean; what is the problem and what did you do to solve this problem) are given here. The authors explained the purpose of the study, but I think, this part of the paper should be rephrased.
Please start with a brief description of the problem, in this context low biosecurity system of the backyard production.
This sentence should be followed with the importance of the molecular detection of blood parasites and supporting the results with microscopic examination.
After, the purpose of the study should be described. For instance, “Therefore, in this study, we aimed to examine the presence of Plasmodium, Leucocytozoon and Trypanosoma in backyard chickens raised in three provinces in Southern Thailand (Nakhon Si Thammarat, Phatthalung and Surat Thani). In this way, the bridge between the problem and the study aim would be stronger.
Thank you for your suggestion. The paragraph was reviewed.
Materials and Methods
Line 113: Microscopical -> Microscopic
Line 137: follow -> followed
Line 154: were -> were
Line 182: R programming -> R
Corrected.
Results
Lines 185-187: “During June 2021 and June 2022, 57 backyard chickens (Gallus gallus domesticus) had their blood investigated for the presence of Plasmodium, Leucocytozoon and Trypanosoma infections.” - > A total of 57 backyard chickens’ (Gallus gallus domesticus) blood samples were investigated to detect the presence of Plasmodium, Leucocytozoon and Trypanosoma infections.”
Corrected.
Lines 187-189: “This included native chickens (n = 33), hybrid chickens (n = 10), laying hens (n = 5) and fighting roosters (n = 9).” Did you find any difference among these groups?
The number of each breed is not suitable for statistical comparison. There is the limitation of project period, sequencing cost and permission of local farmers. Thus, we could not collect more samples. However, we compare the prevalence between areas, using Fisher’s exact test.
Line 218: Microscopical -> Microscopic
Corrected.
Discussion
Lines 322-324: “This study showed a high molecular prevalence of avian malaria (P. gallinaceum and P. juxtanucelare) and leucocytozoonosis (Leucocytozoon spp.) in Nakhon Si Thammarat, Phatthalung and Surat Thani provinces in the Southern of Thailand”. Please rephrase this sentence.
Reviewed.
Line 308: I think, this paragraph should start with the importance of molecular methods. Such as sentence can be added at the beginning of the paragraph “The rapid development of DNA-based molecular methods allows us to understand the species' taxonomy, genetic diversity, population genetic structure, conservation genetics, and parasites”.
Also, these references can be added here (doi: doi.org/10.3390/pathogens10020103, 10.3390/biology12030401, 10.3390/ani10081441, 10.3390/pathogens12050712). After that “In this context, the nested-PCR used detected a higher number of Plasmodium spp., Leucocytozoon spp. and Trypanosoma spp. than the buffy-coat smear…”
Thank you. We include these references (line 398-399)
Conclusion
The conclusion should be rephrased as follows; please start with a brief description of the study (the aim of the study with a sentence), explain the main findings of the study briefly (the results that the authors found), explain how your results contribute to healthy backyard production etc. with 2-3 sentences, explain the limitations of the study and describe the future remarks briefly.
The conclusion was reviewed.
Round 2
Reviewer 3 Report
I am satisfied with revised version
Reviewer 4 Report
The authors improved the manuscript with the previous comments.